# The Fertility of a Concept: A Bibliometric Review of Human Flourishing

**DOI:** 10.3390/ijerph19052586

**Published:** 2022-02-23

**Authors:** Manuel Cebral-Loureda, Enrique Tamés-Muñoz, Alberto Hernández-Baqueiro

**Affiliations:** 1Tecnologico de Monterrey, Department of Humanistic Studies, Monterrey 64849, Mexico; 2Tecnologico de Monterrey, Human Flourishing Projects, Monterrey 64849, Mexico; 3Tecnologico de Monterrey, Department of Humanistic Studies, Ciudad de México 14380, Mexico; albherna@tec.mx

**Keywords:** human flourishing, bibliometrics, Scopus, data mining, cluster analysis, network analysis, virtue

## Abstract

Human flourishing is a thriving concept, whose use has greatly increased among academic researchers from a variety of fields, from the arts and humanities and psychology to the social and environmental sciences and economics. To better understand the concept’s success, this work proposes a bibliometric review, in which statistical methods and data mining were used to analyze 1829 documents, chosen from the Scopus database by searching the term “human flourishing”. Through cluster and network analyses, the study shows the concept’s evolution and composition, as well as its current tensions and trends, in which the predominantly psychological approach is being compensated with social concerns and the search for justice. Furthermore, the concept’s strong philosophical roots provide it with abstract richness and great fertility, which can be seen in keywords, such as virtue or eudaimonia. This bibliometric review proved to be useful for this type of study, despite the limitations imposed by the characteristics of the Scopus database itself.

## 1. Introduction

The question of how well-being and happiness are achieved has been a constant concern for different cultures and societies throughout history [1]. How to learn to live well is a question that has echoed on the shelves of human wisdom for centuries: is it an idea that is understood, is it practiced, is it felt, is it intuited, is it acquired by training? Once the conditions for happiness are met, how are they, in turn, taught? Is it enough to display them? Are they passed on by indoctrination, taught by example, learned by intuition, or genetically transmitted?

In recent years, a concept has emerged, mostly within Western Culture, to answer those questions: the concept of human flourishing. Through this concept, many authors are attempting not only to provide a definition of happiness but also to establish solid, quantitative metrics-supported references to promote the positive development of both individuals and society [2,3,4]. The concept’s broad scope has enabled it to exert influence on many areas: from psychology and mental health studies [5] to social sciences, including studies about work [6,7] and property [8]; from humanistic interests, such as philosophy, ethics, or axiology [9] to more economic approaches impacting business, finance, and management [10]; from the concern with education, public health, or nursing [11] to sustainability and the environmental sciences [12].

In addition, various prestigious academic institutions have created programs and labs focusing specifically on human flourishing, for example, the *Human Flourishing Program* at Harvard University, the Positive Psychology Center at the University of Pennsylvania, the Center for Healthy Minds at the University of Wisconsin–Madison, the Greater Good Science Center at the University of California, and the Center for Positive Organizations at The University of Michigan, to name a few.

The purpose of this research is to gain deeper insight into the formation and recent development of the human flourishing concept, by trying to understand both its origin and the reason for its success. To accomplish this, the study will conduct a conceptual disaggregation to explore the tensions contained in the concept and the most prevalent ways in which it is approached, as well as the trends that might dominate academic research in the coming years. In order to obtain a broad overview of the phenomena, the study opted for a bibliometric review, in which statistics and data mining were used to analyze the massive amounts of data found in the literature.

We found just three previous similar works: Dominko and Verbič [13], who analyzed many words together, related with subjective well-being and happiness, and specifically in the field of economics; Fabricio et al. [14], who studied the term flourishing, but in relation with work contexts and selecting a much smaller dataset; Rusk and Waters [15], who bibliometrically studied the use of the term positive psychology; meanwhile, the current study is specifically about the conceptual use of human flourishing, with a large dataset, and emphasizing the transdisciplinary view.

## 2. Materials and Methods

Bibliometrics is a type of literature review that draws on quantitative and statistical techniques to analyze a large number of documents. Its use is recommended to reduce some of the most common weaknesses of traditional information retrieval approaches, such as ranking biases for relevant documents or resorting to limited textual query modes [16]. The use of bibliometrics in academic research has seen a great increase in recent years, as it is supported by newer and more accessible statistical and computational tools [17]. At a time where academic production is growing exponentially, bibliometrics is becoming quite a suitable tool to overcome the limitations of traditional literature reviews [18].

The current study used data collected from the Scopus academic citation database, which is the most recommended for quantitative analysis [19]; the collection was performed in September 2021. To gather all the data related directly to the concept of human flourishing, we conducted an initial search using the full expression “human flourishing”; the term was written expressly inside quotation marks to force both words to appear together, and the title, abstract, or keywords were selected as possible fields. The search yielded a set of 1144 documents, all of which, after revision, looked like they met the research query adequately. However, there was an important loophole in this first search as there is an alternative expression often used instead of human flourishing, the term “flourishing” alone, which some authors prefer to use. The problem with this term is that it appears in many abstracts—a total of 10,850 in the Scopus database—which are not necessarily related to human flourishing. Most of them belong to the social sciences field, studying, for example, the flourishing of a civilization. To sift out these documents, the term flourishing was entered in the keywords field alone. Under these new settings, only 685 more documents appeared, which, having undergone checking, were incorporated into the database. Thus, as can be seen in Table 1, a total of 1829 documents were finally used for the analysis.

The research was carried out through the use of programming tools to explore and compute data. The analysis was performed using primarily R programming as well as the following packages: bibliometrix [20], useful for many analyses and visualizations; widyr [21], helpful to calculate the correlations between keywords; tidyverse [22], used to manage and visualize most of the data. In addition, to develop the relationship networks, the data was exported to the Gephi software [23] in order to apply network algorithms and enable visualization.

More specifically, three different procedures were applied. First, an exploratory analysis of general counts and observations over the whole data was conducted to identify trends and patterns. Second, a thematic longitudinal evolution map was drawn, using the thematicEvolution function of the bibliometrix package, consistent with a method suggested by Cobo et al. [24] to identify and visualize conceptual subdomains in the scientific production. The function monitors the co-occurrence of two or more keywords in the same document, measures them through the equivalence index [25], and then clusters them. The resulting clusters are plotted in a Cartesian coordinate system, creating four spaces following Callon’s centrality, which provides the degree of cluster interaction with other clusters in the x axis, and Callon’s density, which provides the internal strength of the cluster in the y axis [24].

Finally, considering the sample as a whole and without taking its longitudinal evolution into account, the study conducted a network analysis of the 250 most frequent keywords. The latter was performed using the pairwise_count function of the widyr R package, which was useful in counting the total number of co-occurrences for each pair of keywords. After that, a network was deployed using the Gephi software, which enabled the calculation of their modularity class [26] as well as other network measures such as the betweenness centrality [27], highly useful for a deeper interpretation of network relations.

To obtain a more accurate analysis, some operations were previously processed. Most of them will be explained specifically before each plot or analysis; however, there are a few preliminary decisions worth pointing out. First of all, to compute the keywords, the Authors’ Keyword variable was preferred over the keywords produced by the Scopus database, the reason being that the former was more aligned with what the authors wanted to express as they provided even broader and more varied vocabulary, free from the constraints of predetermined categories.

In addition, the bibliometrix R package offered the possibility of running the keywords through the Porter Stemmer, a significantly useful process when obtaining common terms since it eliminates the small divergences resulting when authors employ variations of a common concept. Nonetheless, Porter stemming was not always enough, and at times, it was necessary to perform some manual modifications. The criteria was to select the most synthetic expression in order to improve management and visualization. It is important to leave a record of such alterations: *human flourishing* was substituted for flourishing, wellbeing for well-being, eudaimonism and some other variations of the term for eudaimonia, capabilities approach for capabilities, and names such as Martha Nussbaum or Thomas Aquinas for their simpler versions, Nussbaum and Aquinas, respectively.

## 3. Results

This section will present the results of the following procedures: (1) a general overview of the data’s recent historical evolution to understand and describe the main trends; (2) a clusterization based on previous observations, focusing more in depth on the historical changes and their relations; (3) network metrics and more specific clusterizations based on semantic features and theoretical references of the field. In addition to the main visualizations given throughout the section, more specific metrics will be provided by adding analytic nuances.

### 3.1. An Exponential Trend since the Early 2000s

Within the Scopus database, the term human flourishing was used for the first time in 1969, in the article Human Flourishing: On the Scope of Moral Inquiry [28]. This article is classified under the arts and humanities field, which from an ethical and even religious point of view, interacts with psychiatry, as it poses the question of what humans need to flourish. The term appeared only a few times during the next 25 years, until 1994, when there were only seven documents featuring it in Scopus.

The first significant growth in the use of the term can be observed during the late 1990s. Between 1995 and 2000 alone, 27 more documents appeared, which represented almost four times more documents in those last 5 years than in the previous 25. Figure 1 shows the exponentially rising phenomenon observed during the following 20 years, highlighting some peaks or incremental points for the periods when the total academic production around human flourishing occurred the most. The first spike happened between 2004 and 2005, when the number of documents published in Scopus went from 11 to 24. A second rise took place between 2008 and 2009, when publications increased from 35 to 62. Similar significant growth spurs took place in 2013, 2016, and 2018.

As can be seen in Figure 1, such evolution in academic productivity created a curve that is more clearly appreciated in the total document count, but also in more specific subject areas. The fields of social sciences, first, and the arts and humanities, next, were the two areas where most of the document production occurred and which also maintained an important level of citations. In fact, some of the peaks previously identified took place in these subject areas, the most important occurring in 2005, when productivity in the arts and humanities increased from 5 to 16 documents, and from 5 to 10 documents in the social sciences.

A closer examination of the documents from these two most productive subject areas reveals the most cited documents in the social sciences to be: Flourishing Across Europe: Application of a New Conceptual Framework for Defining Well-Being [29], with 524 citations; followed *by* A Participatory Inquiry Paradigm [30], with 510 citations, and Introduction: Contributions to the Discipline of Positive Organizational Scholarship [31], with 191 citations. These three articles, written by different authors, claim the benefit of focusing on subjective experiences to measure realities as different, as society as a whole, research epistemologies, and organizations, respectively.

Among the documents classified under the arts and humanities, the most frequently cited are: the book *Why Things Matter to People: Social Science, Values and Ethical Life* [32], with 542 citations; again, the article by Huppert and So [29]—also classified as social sciences—with 524 citations, and the article *Mental health in adolescence: Is America’s youth flourishing?* [33], with 311 citations. Although these documents also elaborate on the importance of subjective experience, a comparison among them will show a focus on the impact that flourishing in daily human lives has as a source of dignity.

Although featuring a lower number of publications than the social sciences and the arts and humanities, the field of psychology can be considered as having a greater impact due to its number of citations. This trend began with Barbara Fredrickson’s article The role of positive emotions in positive psychology: The broaden-and-build theory of positive emotions [34], with 6412 citations at the time of data collection. The article is by far the most frequently cited document of the sample. Opposing traditional psychology, which used to prioritize the study of mental diseases, traumas, and deviations, Fredrickson argued in 2001 that there was important empirical evidence of the effect of positive emotions on all aspects of human life, to the extent of providing strength and resilience in the face of adversity.

The relevance of psychology in the number of citations is constant throughout the sample, as this field contains six of the ten most cited documents: [35] with 1269 citations; again, Fredrickson, but now with Losada [5], with 1182 citations; [36] with 879 citations; [29] with 524 citations; again, Keyes [33], with 311 citations. Some of these articles, however, are also classified under the social sciences and/or the arts and humanities fields. Following the approach first used by Fredrickson, all of the most cited psychology documents develop the idea of a change in perspective, proposing a focus on positive aspects of mental health to achieve individual and collective well-being.

After psychology, but thematically close, medicine is the next subject area in number of documents and citations. Its most cited articles use the term flourishing to refer to mental health, in relation to desirable characteristics in a population [37,38], as well as developing quantitative studies, which use metrics, such as the Flourishing scale type [39].

Finally, it is very important to note the variety of subject areas that make use of the concept of human flourishing. Although with fewer academic documents published, a constant and long use of the term can be observed, since the year 2000, in areas like business and management, nursing, economics and finance, and environmental sciences. The relation of these four subject areas with a concept born in between the arts and humanities and psychology is quite unexpected. Thus, by reaching such diverse disciplines, the concept of human flourishing is proving to be transdisciplinarily fertile.

### 3.2. The Dissemination of the Concept

For a deeper study of the evolution and internal composition of the human flourishing concept, a longitudinal analysis was chosen. Based on the above findings, three time periods were established: the first one starts with the first document of the sample, published in 1969, and covers up until 2004, when the exponential trend began; the second one captures the consolidation of the exponential trend, from 2005 to 2015; the last period tracks the academic production during the last 5 years. The data was processed using the thematicEvolution function of the bibliometrix package, which filtered the 250 most frequent keywords, with a minimum of 5/1000 frequency of a keyword to appear within a cluster. The results are plotted in Figure 2, where the clusters are located in a Cartesian coordinate system, with Callon’s centrality as the x axis and Callon’s density as the y axis. Each cluster is labelled with its three most important keywords, along with the frequency that each keyword reached; however, there were other keywords which were not depicted, due to display limitations.

Figure 2 also shows three different configurations for each period. Between 1969 and 2004, only three clusters emerged. The first one is represented by the keyword flourishing, with high centrality and high density, and it condenses most of the keywords in that period. It also contains the keyword virtue and is very close to another cluster, represented by ethics. Because of their high centrality and density, these two clusters are located at the motor themes zone. A third cluster is identified, equally comprised by liberalism, perfectionism, and utilitarianism, featuring a slightly lower centrality and density, and fully located in the basic themes zone.

Within the second period, it is possible to identify a high number of clusters. The one labeled as flourishing is seen to have greatly decreased in density, while only slightly in centrality, which now places it in the basic themes zone. It is important to note that the rise in the number of documents analyzed during this period is also consistent with the variety of keywords contained in the clusters. In the case of flourishing, the presence of keywords, such as well-being and happiness, stands out; this offers an updated view of the term, which is now more concerned with the subjective perception of happiness and mental health, albeit in an increasingly individualistic way (keywords not plotted but appearing together within the cluster are: positive psychology, subjective well-being, mental health, autonomy, resilience, health, and individualism).

Also within the basic themes, there is another cluster closely accompanying the one labeled as flourishing in this second period. It reveals the classical ethical view, previously identified in the same cluster as flourishing. Labeled under the keyword virtue, this cluster contains many keywords that belong to Aristotelian philosophy, for example, eudaimonia, among others (keywords not plotted but appearing together within the cluster are: ethics, justice, sustainability, practical wisdom, work, politics, and recognition). Another keyword therein is sustainability, which can be interpreted as an update of the Aristotelian concern with immanent justice.

However, within this second period, a new cluster appears in the motor themes zone, under the name of philosopher Martha Nussbaum. The cluster is located in a privileged place, with significantly high density and centrality, which makes it the clearest motor theme for the period. The cluster contains keywords, such as equality and enhancement, as well as others such as liberalism, previously identified in the first period (keywords not plotted but appearing together within the cluster are: identity, institutions, culture, dignity, human capabilities, liberalism, and political). Indeed, this cluster holds multiple semantic connections with the one seeming to be, albeit partly, the motor theme (the capabilities cluster, which also contains the words religion and economics).

Another cluster surfaces between the motor and the niche themes zones, comprised by the keywords emotion, positive emotion, and meaning. It is important to note that, although not apparently seen, the cluster also contains the keyword neuroscience, which is part of this motor, or even niche, theme—as it is placed in the middle—and which takes into account how to achieve motivation and meaning in life. In fact, the term is quite close to the education cluster, which is fully situated in the niche themes and contains keywords associated with common experiences and contemporary claims, allegedly needed to flourish as a society (keywords not plotted but appearing together within the cluster are: Islam, democracy, feminism, and love).

The third period features fewer but larger clusters, where the number of documents with the keywords flourishing or human flourishing rises sharply, increasing from 6 in the first period to 191 in the second and 508 in the last one. The cluster labeled as flourishing recovered a fraction of its density and centrality relative values, although it remained within the basic themes zone. Along with flourishing, other terms, such as well-being and positive psychology, among others, can also be seen (keywords not plotted but appearing together within the cluster are: mental health, depression, psychological well-being, resilience, subjective well-being, mindfulness, and COVID-19). Most of these keywords remain from the second period, keeping their high centrality through the third one and showing an increasingly frequent correlation with flourishing and positive psychology.

Another cluster within the third period is the one equally labeled as religion, sustainability and values, which is also located in the basic themes zone, with high centrality (keywords not plotted but appearing together within the cluster are: human rights, leadership, social justice, and spirituality). Of course, these keywords appeal to different issues, but all of them express claims related to the ethical framework of human flourishing, understanding religion in relation to spirituality, which constitutes a less doctrinal demand of the experience.

At the center of the motor themes zone is the word virtue, with practically the same keywords identified as basic themes in the second period. The cluster grows from 40 keywords to 55 and contains mainly terms from Aristotelian ethical philosophy (keywords not plotted but appearing together within the cluster are: capabilities, eudaimonia, education, disability, health, justice, and meaning). Nussbaum’s capabilities approach is also found within this cluster, maintaining its presence in the motor themes zone, but losing its role as a cluster representative.

Although there are no clusters found entirely in the niche themes zone during the third period, two cross over, partially. The most important one is the cluster labeled as happiness, featuring high density and referring to terms such as life satisfaction (keywords not plotted but appearing together within the cluster are: public health, psychology, optimism, assessment, intervention, policy, and purpose in life). Despite its allegedly highly subjective approach, it also includes keywords such as public health, intervention, and policy, which help the cluster to be understood in a more social and committed way.

### 3.3. Human Flourishing Keyword Network Relations

Having analyzed in detail the evolution of the use of the concept, this study will focus on the network correlations as a whole, much like a snapshot, enabling a view at a glance of all the connections between keywords. With that purpose, a network analysis was performed, clustering the keywords through the modularity algorithm, and taking into account their frequency of co-occurrence as the weight of the edges. Figure 3 shows the resulting network after removing all keywords with a degree below 50.

In the center, there is a cluster in the shape of a circular node, which features both the highest node population (22) and the widest and strongest connections. It contains the two most frequent keywords in the whole data sample: flourishing and well-being. Along with them, there are keywords, such as happiness, positive psychology, mental health, psychological well-being, mindfulness, and life satisfaction. From this set of connections, it can be affirmed that the psychological approach is the closest to the human flourishing concept, highlighting, at the same time, keywords that emphasize individual and internal aspects of well-being and life satisfaction, for example mindfulness, emotion or positive emotion, motivation, resilience, and, clearly, subjective well-being.

It is quite interesting to note that, while precisely, the terms that imply the most negative emotions in the whole dataset (anxiety and depression) are strongly connected with a few of the terms in the main cluster (specifically flourishing, well-being, and mental health), they are identified as a separate, tiny cluster with hexagonal nodes—made up only by these two nodes—and located in the middle of all the positive terminology from psychology.

Second to flourishing and well-being is the pentagonal cluster, made up of 14 modules and featuring virtue as its main node. Virtue is the second most important keyword in the network and appears in strong connection with ethics, another highly relevant key term therein. Without a doubt, the cluster alludes to Aristotelian philosophy, as it contains terms like Aristotle, wisdom, practical wisdom, and common good; it also points to the ideas of other philosophers, such as Thomas Aquinas, which makes the relation with keywords, such as theology, quite logical. Within the cluster, this precise term is related to love, a highly atemporal and universal reference, which, in terms of semantics, can easily connect with many other clusters. Finally, although it belongs to the Aristotelian cluster, justice is another keyword that is also highly connected with the triangular nodes cluster in the network, which features most of the keywords from the capabilities approach.

Effectively, the triangular nodes cluster, made of seven nodes, represents the constellation of the capabilities approach, as it contains terms such as social justice, human rights, equality, and human dignity. It is important to note that, unlike the flourishing cluster, which is dominated by internal and subjective perspectives—highly characteristic of the positive psychology movement—these triangular nodes refer to highly social and external qualities. This fact confirms that the capabilities approach plays a complementary role in the academic literature on human flourishing, addressing social and mostly external issues, which are barely present within the psychology cluster.

Finally, the square node cluster, consisting of 12 nodes, is quite extensive and spread out throughout the network. Its main nodes are education, meaning, religion, values, and community. These keywords mediate between the more internal concern of the psychology area, through keywords, such as meaning, values, spirituality, identity, or autonomy, and the more external and ethical common issues addressed by the capabilities approach, such as education, community, disability, culture, and morality.

### 3.4. The Pathways of the Human Flourishing Network

The betweenness centrality is an alternative measure to understand a network, not just by its terms’ frequency of appearance, but by the way in which more nodes connect better and faster. In the case of this human flourishing network, they will be the keywords and approaches that can connect different trends, playing the important role of mediators within the whole database.

Table 2 shows the network’s top twenty keywords, ranked by their betweenness centrality. For the first three nodes (flourishing, well-being, and virtue) the betweenness measure matches the weighted degree, which is normal because the last one expresses the nodes’ gross frequency within the network. However, after these first three places, some interesting movements can be observed.

The first keyword whose weighted degree stands out is eudaimonia. Although it features a lower absolute frequency of appearance than that of keywords such as happiness, positive psychology, Aristotle, or mental health, this word obtains a very high position when ranked by its betweenness centrality. As is well known, eudaimonia is a term of Greek origin, introduced by Aristotle, yet as can be seen from the analysis, its capacity to be used in different contexts surpasses this strict context; despite its being one of the capital terms in the field of ethics, eudaimonia was identified as belonging to the main cluster and being well connected with keywords, such as practical wisdom, happiness, or flourishing, but also as reaching a wide variety of terms, such as health, sustainability, work, depression, spirituality, environmental ethics, and liberalism.

Another keyword whose weighted degree position also stands out is ethics, which as is the case with eudaimonia, comes from a classical philosophical approach, yet connects truly well with most individualistic terms of the network, such as subjective well-being, happiness, life satisfaction, or spirituality, and with the most common and external concerns, such as economics, education, disability, justice and social justice, human rights, and common good, among others. Furthermore, ethics is highly connected to certain keywords, such as mental health, quite a specialized and specific term that does not bear connection with many others.

Precisely by looking at the opposite phenomena, that is, at the terms that have a high weighted degree but a lower betweenness centrality than expected, other valuable observations can be made. The first one is that, although positive psychology has more connections and even more betweenness centrality than capabilities, proportionally the last one is more efficient when connecting different keywords. Along with positive psychology, other keywords with an important difference between these two values are mental health, happiness, and, more surprisingly, anxiety and depression. All these relations denote keywords used in specific fields that are not commonly used in research around human flourishing, so they will never be representative of the whole network.

## 4. Discussion

There are a few questions, and even tensions, that surface after the described analysis, especially those that allow for a deeper analysis of the configuration and composition of the concept, which should be explicitly asked in order to pose the challenges that will affect this field of research in the coming years.

The main tension has to do with the individualistic drift of the human flourishing concept. Mostly influenced by the psychological point of view, a number of concerns about subjective well-being can create an obsessive worry about one’s own happiness, forgetting many of the issues that society faces as a whole. Unfortunately, such an individualistic approach has been used for marketing and commercial purposes, which distorts scientific knowledge. In this regard, Cabanas and Illouz [40] popularized the expressions “the industry of happiness” or “imperative happiness”, criticizing the imposition of the idea of well-being to control people’s lives.

Of course, this trend is also visible in the analysis. For example, indicators like the growing importance that terms, such as happiness, life satisfaction, positive psychology, and positive emotion, display in the thematic evolution map (Figure 2), or the high number of positive terms that they are closely related to in the biggest, and central, cluster of the final network (Figure 3), or the isolation that terms like anxiety or depression acquire in this final network, all confirm the growing importance of the subjective and individualistic approach within the human flourishing field of research.

However, there is also evidence in the current analysis which denies or restricts this interpretation. First is the relevance that concepts like sustainability, human rights, public health, education, disability, and justice acquire within relevant clusters, identified in the last period of the thematic evolution map (Figure 2). Indeed, public health appears under the cluster labelled as happiness and life satisfaction, thus, showing that the term implies a wider view than that centered on the individual alone. Another keyword that complements this individualistic bias is environmental ethics, identified in the final network, and joined also by sustainability (Figure 3).

The second important aspect to be noted is the importance of the approaches related to Martha Nussbaum and the theories of justice, which constantly bring the concept of human flourishing back into a more public debate, vindicating human rights and dignity for all human beings. Such importance is also reflected in the relevance that these concepts have within the network along with others, such as disability, common good, community, equality, and social justice (Figure 3).

Finally, it can also be argued that the most frequently cited papers, previously included in the exploratory section, directly address social issues from a point of view that clearly surpasses the individual. Explicitly, the article by [29] argues that well-being must be evaluated not only in terms of life satisfaction, but also as a multi-dimensional construction, where social and cultural aspects have to be taken into account. Ruger [41], on the other hand, incardinates the concept of human flourishing in quite an external and common place: the intersection of ethics, economics, political science, law, and human rights. Most of these authors, including Keyes [33,36], argue that good public and social institutions are essential for the proper achievement of human flourishing.

Of course, despite all these nuances that condition the drift of human flourishing as a concept too oriented to subjective goals, it cannot be denied that this trend exists and that its dependence on just one author or school of thought, such as Martha Nussbaum and the capabilities approach, could represent a serious danger for the future power and fertility of the concept. In fact, the common goals that appear in the second period as niche themes, under the label of education, partially disappear in the last period or are reabsorbed under the keyword virtue, thus, losing most of their specific approach. This, indeed, could signal the beginning of a trend.

Returning to the Scopus database, other documents were found which issued a warning against the concept of human flourishing becoming entirely individualistic: Newton et al. [42] appealed to a broader conception of land and property for human flourishing, beyond individualism; Martin [43] advocated the historical and sociocultural resources that constitute self, beyond the individualistic conception; Becker and Marecek [44] issued a call to go beyond individual isolation towards social environment concerns and activism; Christopher et al. [45] understood the overcoming of individualism as a challenge from inside positive psychology itself; Taylor-Sands [46] addressed bioethical concerns; Michel [47] criticized modern psychiatry’s hidden and limitative understanding of human flourishing and pointed to the need to cater comprehensively for the unity of a life; Koch [48] wrote in the context of an enthusiasm for technocracy and human enhancement; Evans [49] or Hwang [50] discussed religious practices in general; Nelson and Slife [51] contrasted psychological meaning specifically with Christian thought; and Holland [52] vindicated a catholic conception of human flourishing, away from the neoliberal imperatives.

It is worth pointing out that most of these documents do not deny the opportunity of the concept of human flourishing; rather, they are trying to improve, nuance, or extend some of its uses and meanings. Meanwhile, there are also a few works that specifically reject such criticisms as not appropriate, for example, Godfrey-Wood and Mamani-Vargas [53], who explore the relationship between collective and individual capabilities, or Skorupski [54], who justifies the social concern with individuals’ good.

## 5. Conclusions

Human flourishing is quite a thriving concept, showing an exponentially growing trend in academic production, which has not ceased to increase in recent years. Although the concept was born in the area of the arts and humanities, it did not truly take off until the year 2000, when it began to be used in the field of psychology. Both Fredrickson’s proposal [34], and the ensuing use by other positive psychologists, lays much of the groundwork for its development. From then on, interaction among the arts and humanities, social sciences, and psychology resulted in most of the research production on the concept, with a special focus on happiness, well-being, virtue, health, and life satisfaction. These terms, which characterize the positive psychology approach quite well, have finally been found to be part of the most important terminology in the conception of human flourishing.

However, it cannot be said that the concept of human flourishing is the legacy of any single discipline, but rather a transdisciplinary concept, with the capacity to be fruitful in many different fields, such as the social sciences, business, medicine, nursing, economics and finance, and environmental sciences. Such a proliferation has been interpreted as its fertility, based on the way in which its meaning has been used and enriched in many fields.

Indeed, the network graph shows that, although positive psychology appears most frequently in relation to flourishing, acquiring great centrality in the last two periods, some other terms have been identified with greater intermediation, which are needed to complete the map of human flourishing network relationships.

Among these more interconnected terms, the capabilities approach, mainly developed by Martha Nussbaum and Amartya Sen, stands out. This approach complements that of positive psychology, not only in terms of frequency of appearance, but also conceptually, referring the most to external and common concerns around social justice, human rights, equality, and dignity. Between 2005 and 2015, those keywords comprised the cluster with the greatest centrality and density in the sample.

However, the possibility of individualism in the use of human flourishing has also been addressed in the discussion, where it was shown that this is a limited issue. Firstly, because of the manifest presence in the sample of concepts, such as environmental sustainability, education and public services, or social justice (all very relevant and well-connected keywords in the analyses). In addition, it was also observed that most cited authors in the sample strived to link individual flourishing with social and collective flourishing.

Looking specifically at the scientific production over the last five years, it can be pointed out that the density of the main clusters has grown, creating a map with large clusters, both in the basic and in the motor themes zones. This indicates not just that the concept is being used, but that it is being used in new ways. In particular, in the last five years, the motor cluster is composed of philosophical concepts, as in virtue, ethics, Aristotle’s philosophy, eudaimonia, justice, and meaning. In addition, these are the keywords that were also identified at the top of the betweenness centrality list, showing how the philosophical point of view is specially contributing to mobilize and expand the use of the term transversally, throughout its network of relationships.

The current study has mainly shown the theoretical conformation of the human flourishing concept; however, some practical and managerial observations can be made. Most likely motivated by the abstract and philosophical character that the concept has earned, it has been shown that it is a concept easy to be applied in different practical and professional environments, mainly those where, together with technical requirements, ethical and empathetic considerations are also needed. The amount of articles related with psychology, medicine, nursing, as well as the variety of keywords of this kind that have appeared in the study, support this consideration.

Furthermore, managerial implications have been also detected. Areas like business and management, economics and finance or environmental sciences have already introduced the term in their reflections, and keywords, such as work, housing, education or assessment, appear with high centrality and are well connected in the analysis. The advantage of its use is that human flourishing allows us to think about these questions with psychological and philosophical roots, based on the individual, but also attending to social justice and environmental crises, as demonstrated throughout the analysis.

For further studies, it should be considered that, although Scopus is one of the most relevant databases for academic production, the results that it provides have to be taken with caution [55]. Indeed, recent studies have demonstrated that the more an academic field grows, the more centralized it becomes [56]. Because of this, further analysis should be carried out using alternative approaches. For example, in addition to Web of Science, which is a database quite similar to Scopus, other platforms could be explored, in order to include the literature of other geographical latitudes, gaining intercultural understanding. Altmetrics studies have been used in recent years [57]; this might be suggestive when considering that the concept of human flourishing includes a social component.

## Figures and Tables

**Figure 1 ijerph-19-02586-f001:**
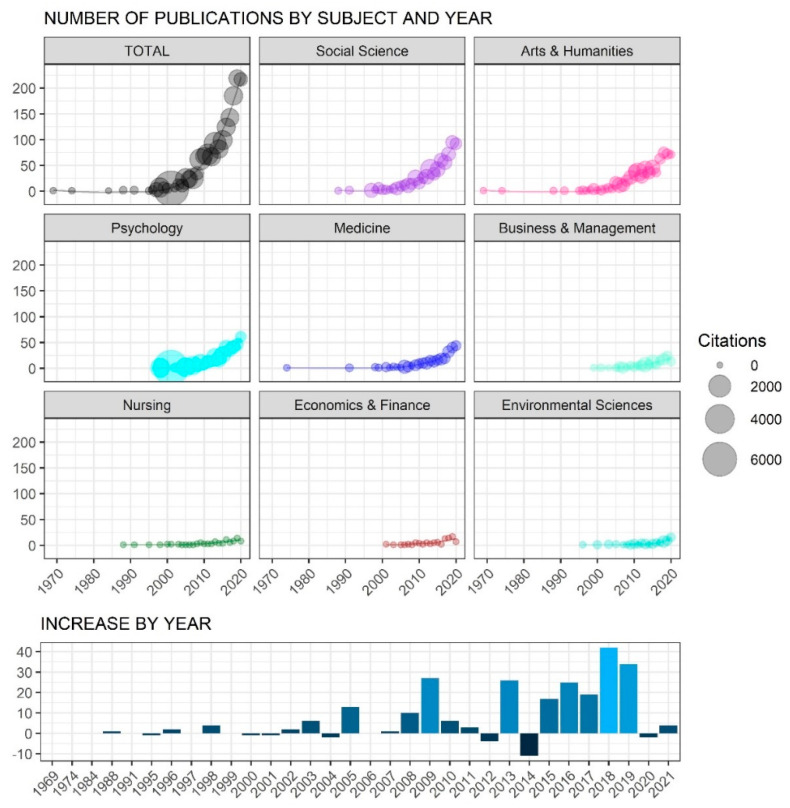
Human flourishing academic productivity, by year, faceted by Scopus subject areas.

**Figure 2 ijerph-19-02586-f002:**
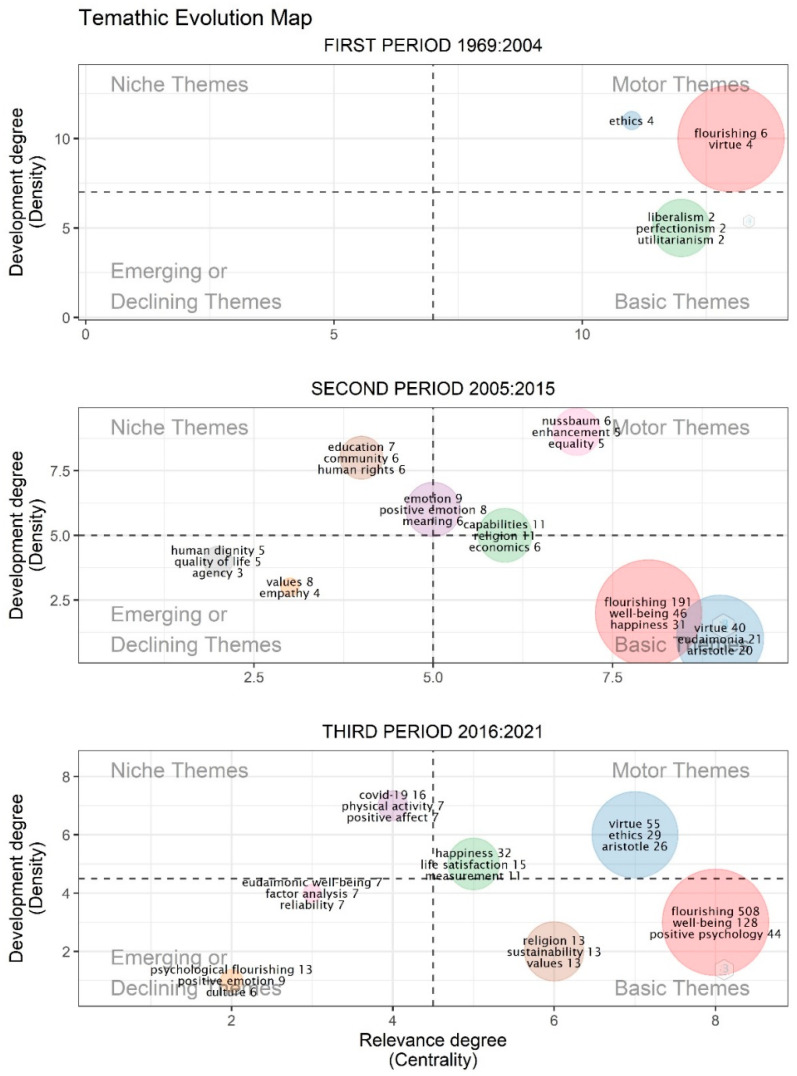
Authors’ Keyword thematic evolution map clusterized items taking into account the 250 most frequent words overall, with a minimum of 5/1000 frequency and deploying the three most valued labels. The size of the clusters is proportional to the volume of keywords it contains.

**Figure 3 ijerph-19-02586-f003:**
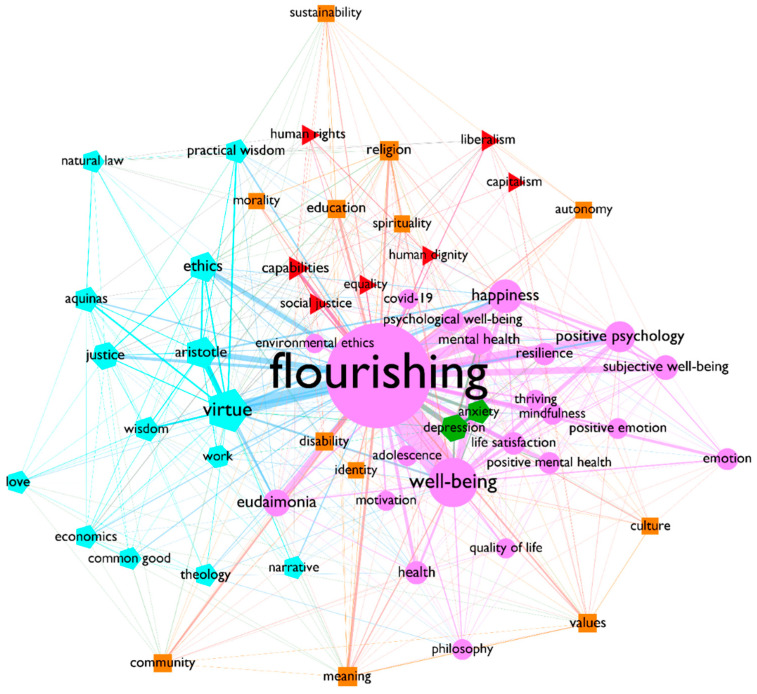
Authors’ keyword network clusterized by their modularity class. The clusters are represented by the different shapes and colors of the nodes; the width of the edges shows the frequency of co-occurrence between a pair of keywords; the size of the nodes is their weighted degree, and the size of the labels represents the betweenness centrality. The network was filtered for keywords with a degree range higher than 50. The plot was drawn with Gephi software. For an interactive version of this Figure, see the supplementary materials (http://oteatroresoante.es/network/, accessed on 15 December 2022).

**Table 1 ijerph-19-02586-t001:** Summary of the data collected from the Scopus database.

Main Information about Data	
Timespan	1969 to 2021
Sources (journals, books, etc.)	1147
Documents	1829
Average citations per document	15.96
Average citations per year & document	1.728
References	98,993
Document Types	
Article	1253
Book chapter	275
Review	112
Book	90
Other (conference papers, notes, etc.)	99
Authors	
Authors	3105
Author appearances	3750
Authors of single-authored documents	884
Authors of multi-authored documents	2221
Author Collaboration	
Single-authored documents	1024
Documents per author	0.589
Authors per document	1.70
Co-Authors per document	2.05
Collaboration index	2.76
Document Contents	
Author’s keywords	4775

Note: The data was collected in September 2021, according to the search TITLE-ABS-KEY (“human flourishing”) OR KEY (flourishing).

**Table 2 ijerph-19-02586-t002:** Top twenty nodes in the authors’ keyword network on human flourishing.

Keyword	Modularity Class	Weighted Degree	Betweenness Centrality	Closeness Centrality	Eccentricity
Flourishing	1	2120	2533.98	0.983607	2
Well-being	1	754	499.48	0.701754	2
Virtue	2	482	450.94	0.693642	2
Eudaimonia	1	212	225.02	0.628272	2
Happiness	1	378	212.17	0.634921	2
Ethics	2	188	196.11	0.615385	2
Positive psychology	1	312	153.13	0.615385	2
Capabilities	3	112	101.03	0.571429	3
Health	1	124	100.04	0.585366	2
Aristotle	2	242	96.84	0.591133	3
Justice	2	126	82.1	0.576923	3
Resilience	1	108	77.01	0.56338	2
Mental health	1	248	70.55	0.566038	3
Education	4	88	69.98	0.566038	3
Values	4	86	69.01	0.56872	2
Subjective well-being	1	162	67.14	0.560748	2
Religion	4	104	59.92	0.56872	2
Meaning	4	106	56.97	0.574163	2
Life satisfaction	1	112	54.52	0.550459	3
Emotion	1	92	44.49	0.555556	2

Note: the rows are ordered according to their betweenness centrality. The weighted degree expresses the total number of connections reaching a node; the closeness measures the average distance from an initial node to all others in the network; eccentricity gives the distance from a node to the furthest one away from it in the network.

## Data Availability

The data are available at figshare, with the DOI https://doi.org/10.6084/m9.figshare.16896643.v1.

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
