# Peer review of "The Fertility of a Concept: A Bibliometric Review of Human Flourishing"

_ijerph, 2022, doi:10.3390/ijerph19052586_

Round 1
Reviewer 1 Report
I consider it to be a well-designed and structured bibliometric study. The results are easy to read and it offers novel and important insights for researchers on Human
Flourishing.
For this reason, I strongly consider that this manuscript should be published.
It would only improve to add some comment, in the discussion to compare in more detail their findings with respect to other studies.
Author Response
Regarding to the comment "improve to add some comment, in the discussion to compare in more detail their findings with respect to other studies", because of the limits of the extension, instead of referring it in the discussion, the question was retaken in conclusions, 5th paragraph.
Reviewer 2 Report
Congratulations on your research. The theme is important and brings a contribution to the literature.
I have a few suggestions:
1 - Faced with so many other databases, why was only the SCOPUS database selected? Justify the choice of only one database.
2 - Given the robustness of the analyses, I suggest highlighting the main contributions of the article, as well as the theoretical, practical, and managerial implications.
Author Response
About Scopus as unique database, it has been added the following comment and reference [second section Materials and Methods, 2nd paragraph]:
The current study used data collected from the Scopus academic citation database, which is the most recommended for quantitative analysis [19]; the collection was performanced in September 2021.
Here the reference that justifies the selection:
Baas, J.; Schotten, M.; Plume, A.; Côté, G.; Karimi, R. Scopus as a Curated, High-Quality Bibliometric Data Source for Academic Research in Quantitative Science Studies. Quantitative Science Studies 2020, 1, 377–386, doi:10.1162/qss_a_00019. About the second reviewer comment, we made some adjustments in the conclusions to include the suggestion. Particularly, the conclusions were rewritten in order to specify better, in a more synthetic way, each of the main findings obtained. In addition, a new paragraph was added where practical and managerial applications of the study are commented. For more detail, see conclusions.Reviewer 3 Report
Thanks for a study that touches upon many different disciplines and is particularly interesting to those who appreciate language and its nuances. I do recommend that when discussing the small number of references on p. 4 that you note whether a particular author wrote multiple works. On page 6, first paragraph, line 2, note that "less number" should be replaced with "fewer." I appreciate your research model that takes into account other and similar words for the concept of "human flourishing."
Author Response
The correction, on page 6, first paragraph, line 2, from "less number" to "fewer" was applied.
Regarding the comment "that when discussing the small number of references on p. 4 that you note whether a particular author wrote multiple works", it was pointed out that these specific works were written by different authors; see subsection 3.1. An Exponential Trend since the early 2000s, 4th paragraph.
This manuscript is a resubmission of an earlier submission. The following is a list of the peer review reports and author responses from that submission.